# Explainability in Practice: Estimating Electrification Rates from Mobile Phone Data in Senegal

**Laura State**
University of Pisa
Scuola Normale Superiore
Pisa, Italy
`laura.state@di.unipi.it`

**Hadrien Salat**
Alan Turing Institute
London, UK
`hsalat@turing.ac.uk`

**Stefania Rubrichi**
SENSE
Orange Innovation
Châtillon, France
`stefania.rubrichi@orange.com`

**Zbigniew Smoreda**
SENSE
Orange Innovation
Châtillon, France
`zbigniew.smoreda@orange.com`

## Abstract

Explainable artificial intelligence (XAI) provides explanations for not interpretable machine learning (ML) models. While many technical approaches exist, there is a lack of validation of these techniques on real-world datasets. In this work, we present a use-case of XAI: an ML model which is trained to estimate electrification rates based on mobile phone data in Senegal. The data originate from the Data for Development challenge by Orange in 2014/15. We apply two model-agnostic, local explanation techniques and find that while the model can be verified, it is biased with respect to the population density. We conclude our paper by pointing to the two main challenges we encountered during our work: data processing and model design that might be restricted by currently available XAI methods, and the importance of domain knowledge to interpret explanations.

## 1 Introduction

Explainable AI (XAI) provides techniques to better understand machine learning (ML) models. This is motivated by their lack of transparency, and an increased use of these models in resource allocation problems that critically affect individuals, such as hiring, credit rating or in public administration. [1] While many XAI methods have been proposed, there is a certain lack of work that uses these methods on real-world data and thus confirms their relevance [19].

In this work, we present a use-case of XAI: we train an ML model to estimate electrification rates in Senegal, and evaluate it using two popular XAI techniques. The estimation of such socio-economic indicators can support policy planning and is assumed to be a less costly and time-consuming alternative to traditional approaches such as collecting census or survey data. Policy planning involves considerable amounts of resources, thus, requires transparency and accountability. We draw on a dataset of mobile phone data, collected in 2013 and provided during the Data for Development challenge by Orange in 2014/15. We combine it with extracts from the 2013 census in Senegal, and estimate the electrification rate around single cell tower locations.

The contribution of our work is twofold: first, we show how XAI methods can be used to verify an ML model, and that our model is biased w.r.t. population densities. To verify means in our case

---

[1] `https://algorithmwatch.org/en/automating-society-2020/`

2022 Trustworthy and Socially Responsible Machine Learning (TSRML 2022) co-located with NeurIPS 2022.

to show that the model indeed relies on features that relate to the predicted outcome, as given by domain knowledge. Thus, we confirm the relevance of XAI techniques. Second, we point towards two challenges of deploying XAI in practice that emerged during this work: pipeline design, and domain knowledge.

The paper is structured as follows: section 2 provides the relevant background, section 3 information on the used data, followed by a description of the experiments in section 4. Results are discussed in section 5, followed by the limitations in section 6 and we conclude our paper in section 7. [2]

## 2 Background

### 2.1 Explainable AI

The field of explainable AI can be distinguished along three dimensions: black vs white box approaches, local vs global, and model-agnostic vs model-specific [10]. The term *white box* refers to models that are interpretable, or explainable by design, while a *black box* (BB) model is not interpretable, or accessible, e.g., due to intellectual property rights. The majority of used ML models belong to the latter, and it is exactly for those models that we have to design explanation techniques. These explanations can be distinguished by scale. *Local* techniques explain the prediction for a single data instance (often an individual, in our case a single cell phone tower). Here, most prominent approaches are LIME (Local Interpretable Model-agnostic Explanations) [24] and SHAP (SHapley Additive exPlanations) [15]. Opposed to this, *global* approaches tackle a full explanation of the system, usually by fitting an interpretable surrogate model. One example is the TREPAN algorithm [7], building a single decision tree over a neural network. Last, we differentiate between approaches that work on any (*model-agnostic*) or only one (*model-specific*) model. LIME [24] and SHAP [15] are considered model-agnostic, TREPAN [7] model-specific. A full survey of approaches can be found elsewhere, e.g., [10, 17, 2].

### 2.2 Mobile Phone Data and Electrification

Mobile phones are a rich source of information, providing details about time, length and location of calls and other data. Combined with the fact that mobile phone penetration is generally high, [3] this opens possibilities for research, public policy, infrastructure planning, etc. Predicting socio-economic indicators from remotely accessible data is popular, and we observe an interest in using such data in countries of the Global South, where it might be a less costly and time-consuming alternative to traditional approaches such as collecting census or survey data. Different indicators can be predicted, and they vary based on available data and method. Examples based on mobile phone data are [5, 27, 26] (estimating socio-economic status and welfare indicators, literacy rate, population densities and electric consumption), approaches using additional data sources are [23, 28] (estimating poverty measures). In many cases, these studies are framed within the 17 SDGs. [4]

In this work, we focus on the estimation of electrification rates using mobile phone data. Relevant studies that investigate the relation of electricity and other indicators such as mobile connectivity or volume of visitors in Senegal are [12] and [25].

## 3 Dataset

### 3.1 Mobile Phone Data

We use mobile phone data provided in the form of pre-aggregated call detail records (CDR), which were made available during the second Data for Development challenge launched by Orange in 2014/15. The original data were collected by Sonatel (*Société Nationale des Télécommunications du Sénégal*), who is the leading telecommunication company in Senegal (market share of 65% in 2013). CDRs are generated for billing purposes, and are proprietary.

---

[2]The code of the project can be found here: `https://github.com/lstate/explainability-in-practice.git`.

[3]`https://ourworldindata.org/grapher/mobile-cellular-subscriptions-per-100-people`

[4]`https://sdgs.un.org/goals`

Table 1: Features and abbreviations. Events type (number of calls, call length or number of text messages) will be indicated by subscripts (CN, CL, SN).

| | |
|---|---|
| te | number of events *within* Voronoi cell |
| | (synonymously used: total events) |
| out | number of outgoing events |
| in | number of incoming events |
| dc | degree centrality |
| cc | closeness centrality |
| out/in | ratio of outgoing over incoming events |

Senegal is a sub-Saharan country, located on the Northern Hemisphere near the Equator, on the West Coast of Africa. It covers 196,712 km$^2$. In 2013, the population count approached 14M. [5] , the year when the data were collected. The original dataset contains more than 9 million individual mobile phone numbers, with a hourly resolution. Sonatel anonymized the data, Orange pre-aggregated and processed it further. The resulting dataset holds (cell) tower-to-tower activity, for calls (including call length) and text messages separately. Spatial coverage between call and text message data is different, text message data less available in the Eastern part of Senegal (see appendix, section A.1). These areas fall together with areas that are less electrified which might suggest a connection between access to electricity, poverty rate, literacy and text message activity (see [12] for a study on the impact of access to electricity). All details on the dataset can be found here [18].

We process this data as a weighted communication network (CNW, one network per data type) based on the overall activity of 2013. The CNW is directed and uses cell towers as nodes, call and text activity to determine the weights and direction of edges. The cell tower locations are used to map the activity spatially, and we rely on a Voronoi cell tessellation. From each network, we extract 6 features which are listed in table 1, they form the final dataset which has dimensions of $1587 \times 18$, i.e. cells times features. Due to space reasons and the focus of this paper, more details are in the appendix A.2.

## 3.2 Electrification Data

Electrification rate, population count and population density originate from the 2013 census in Senegal [3]. Pre-processing is identical to Salat et al. [25]. The census contains questions about the source of lighting for each household and therefore informs us of stable access to electricity. In the context of this study, stable access means access either to the main power grid or to photo-voltaic systems that benefit from the year-round high solar irradiation in the country. The electrification rate is computed for each Voronoi cell as the ratio between the total number of households with stable access and the total population count, assuming a homogeneous distribution of the population inside each census unit (*Commune*). The study reports that the resulting electrification rates were in good agreement with the fine-grained nighttime lights intensity provided by NOAA [20]. The supporting shapefile containing the geographic boundaries and population counts at commune level was provided alongside another previous study[26].

Figure 1 shows electrification rates in Senegal. Higher rates are observed around major cities: Dakar, the Capital, in the Western most part of the country, Saint-Louis in the North West, and the culturally significant city of Touba in the center. The electrification rates are also high along the more densely populated Northern/Eastern borders with Mauritania. This follows closely the electric grid of Senegal, discussed in [16]. We therefore observe a correlation between areas with high electrification rate and high density of cell towers.

## 4 Experiments

### 4.1 Preparation

Each data row (a Voronoi cell) is labeled by its electrification rate. We bin the data such that it holds 10 classes, ordered by electrification rate, and with a bin size of 0.1 (0..0.1 electrification rate for class 0, 0.1..0.2 electrification rate for class 1, etc.). The resulting distribution is skewed towards

---

[5]`https://data.worldbank.org/country/senegal`

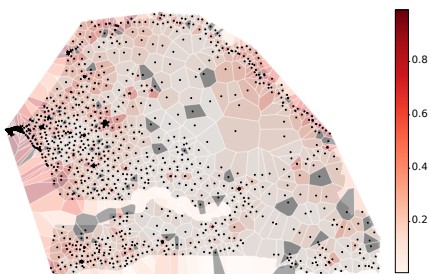

Figure 1: Electrification rate as computed from census data, plotted over the Voronoi cell tessellation.

higher values. [6] We subsample the elements of class 9 (electrification rate 0.9..1), for the training sets only. We sample such that numbers in class 9 are reduced to the average count over all classes. We split the dataset to create a training and a test set for the ML classifiers (7 : 3), and split the test set to create a training and a test set for the explanations (7 : 3).

## 4.2 Models

**Classification** We train several ML standard models to estimate electrification rates: a decision tree (DT), a random forest model (RF) containing 100 decision trees, and an extreme gradient boosting model (XGB) with default parameters. We compute the accuracy, the mean absolute error (MAE) and the ratio between the MAE and its maximum ($MAE_{max} = 9$).

**Explanations** We compute local, model-agnostic explanations based on LIME and SHAP. We do so for the following reasons: both methods, including their pitfalls, are well known in the community, [7] methods are easily accessible, and rely on interpretable features [17]. LIME [24] uses a randomly generated neighborhood, weighted according to a distance and a kernel function, to fit a linear regression around the data instance in focus. This regression approximates the decision boundary, and its weights are interpreted as the *importance* of the features. A positive importance pushes the classifier towards the predicted class, while a negative pulls it away and towards one of the other (here: nine) classes. Therefore, both the sign and magnitude of the importance matter. SHAP [15] explanations are based on a game-theoretic approach, and provide for each feature its *contribution* towards the predicted outcome. Contributions are not only specific to an instance but also to the class. If we sum over all contributions w.r.t. to a class $C$, and add this value on top of the expected value $E_C(f(x))$, we reach the predicted value of our model $f(x)$, which in turn determines the class (the highest value wins). Therefore, the sign and magnitude of a feature contribution matter as well.

We compute explanations only for the best performing model, assuming that only this model would be used in deployment. As LIME needs to compute some basic data features to generate explanations, it has to be initialized on the explanation training set. We use LIME tabular and retrieve the five most important features ($d = 5$). Evaluations of explanations are computed over the explanation test set.

## 4.3 Urban and rural areas

We hypothesize that the ML model predicts electrification rates with different accuracy for rural and urban areas, i.e. that the model is *biased* w.r.t to the population density. Therefore, we identify these regions in the test datasets, and calculate the disaggregated accuracies. Further, we compute the disaggregated explanations. We identify urban regions based on a population density $p > 1000/km^2$, as previously done in [25], otherwise as rural.

---

[6] The ratio between instances in class 9 and the full dataset before subsampling is $imb = 33\%$.

[7] $12.7M$ downloads of LIME python package, $63M$ downloads of SHAP python package, retrieved on 6th of April 2022 `https://pepy.tech`

Table 2: Classification results. The higher the accuracy the better, the lower the $MAE$ the better. Best values are underlined.

| model | $acc$ | $MAE$ | $MAE$ / $MAE_{max}$ | $acc_{urban}$ | $acc_{rural}$ |
|-------|-------|-------|---------------------|---------------|---------------|
| DT | 0.428 | 1.258 | 0.140 | 0.714 | 0.247 |
| RF | 0.516 | 0.972 | 0.108 | 0.854 | 0.301 |
| XGB | 0.491 | 1.027 | 0.114 | 0.789 | 0.301 |

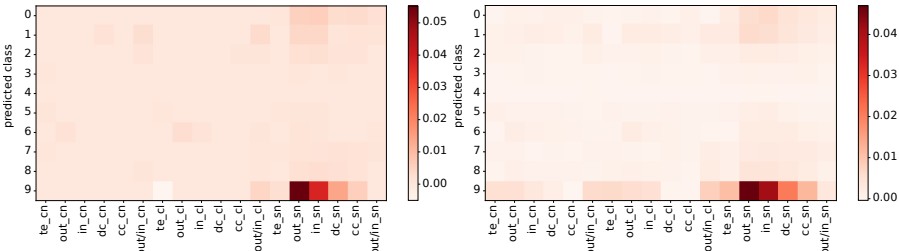

Figure 2: Average importance (left, LIME) or contribution (right, SHAP) of a feature w.r.t. predicted class. Both plots computed for RF classifier. Reminder of notation: table 1.

## 5 Results

### 5.1 Classification

While all models perform similarly well, best performance is achieved by the random forest model ($acc = 0.516, MAE = 0.972$, see table 2). As the classification is ordered, an MAE around one means that on average, the electrification rate is wrongly estimated by $0.1$, which is acceptable. Note that while we need a sufficiently high performance of our models to proceed, the focus of this paper is not on model performance.

### 5.2 Explanations

We display results in figure 2, where we plotted the average *importance* (left, LIME) and the average *contribution* (right, SHAP) w.r.t. the *predicted* class. Both for LIME and SHAP, features based on text message data are highly relevant. This is especially prominent for outgoing and incoming events and the degree centrality, and for class 9. Only in the case of LIME, we observe negative values (e.g., total events based on call length, for class 9). The importance, *and* high contribution of text message data for the prediction is in line with the observation that text message activity could be correlated with the electrification rate (see also section 3.1). As such, information provided by this type of data is highly relevant for the model, providing cues to better discriminate.

For SHAP, we also computed the average over feature contributions w.r.t. all possible output classes. An analysis of this can be found in the appendix (section A.3).

Both methods confirm that the model relies on features that indeed relate to the predicted outcome. While a more in-depth analysis of the model and explanations is needed, this is a first good result.

### 5.3 Urban and rural areas

Disaggregated accuracies are displayed in table 2. While the accuracy for rural areas is always lower than the accuracy as computed for the full test set, the opposite is the case for urban areas. This difference in accuracy means that the model is *biased* w.r.t. to the population density.

We also computed disaggregated explanations, depicted in figure 3. They confirm the general relevance of features based on text message data for the prediction, as demonstrated already in the general case (previous section). Unsurprisingly, they show different distribution of classes: while urban areas belong to class 6 or higher, rural areas span all classes. This is also reflected in the relevance of the features based on text message data, we observe highest importance vales (contributions) as provided by LIME (SHAP) for different classes, depending on whether we look

at urban or rural sub-populations. We further observe that the feature importance values, or feature contributions, are consistently higher in magnitude for the urban compared to rural sub-population. This supports our finding that the model is biased w.r.t. the population density.

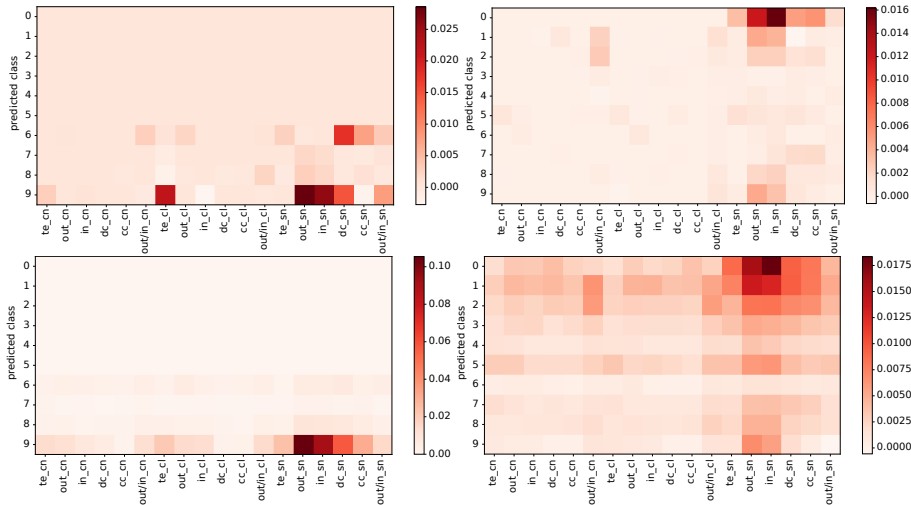

Figure 3: Explanations by sub-population, w.r.t. predicted class: for urban regions (left) and rural regions (right), based on LIME (top row) or SHAP (lower row).

## 6  Limitations

Although the data come from a network provider who is the market leader (65% in 2013), they are not fully representative of the population. A good starting point to investigate this further is the work by Pestre et al. [22]. Salat et al. [25] point towards other biases, for example due to a shift from the mobile phone network to relying increasingly on internet platforms such as Facebook. Accounting for these biases, including further work on the bias w.r.t. population density is a next step. Also, we would like to apply other XAI methods such as LORE [9] to the ML model to understand whether we can extract some additional information, and run the explanations across all trained models.

Data used in this work are proprietary and private. It is connected to individuals (their mobile phone) and to be processed only under their consent. It was provided in an anonymized form and further aggregated to safeguard individuals. Being proprietary, the data cannot be shared beyond the project. An alternative could be relying on other open source data, such as satellite imagery, which have already been proven useful for similar projects. While being fundamentally different to mobile phone data, extending the XAI use case to these data are a valuable path to follow.

The data we used originate from Senegal, a country of the Global South. Being situated in Western Europe, we should critically reflect on how this might perpetuate power relationships [1]. While our focus is on providing a use case of XAI, and to primarily verify a specific ML model that can estimate electrification rates in Senegal, we acknowledge the importance of local knowledge and domain expertise when evaluating data, and specifically when drawing policy implications [1, 14].

## 7  Conclusion

In this work, after showing that electrification rates can be estimated from mobile phone data, we applied two local, model-agnostic explanation tools to verify our model. Both explanations perform well, and agree with each other on stressing the relevance of text message data for the predicted outcome. They confirm the general validity of the model. We also showed that our model is biased w.r.t. to population densities. Thus, areas located in a rural area receive an unfair prediction, i.e. are more likely to be linked to a wrong electrification rate.

While the prediction of socio-economic indicators from remotely accessible data is not novel in itself, it is of high relevance, e.g., to support policy planning. Verifying such a model using XAI techniques

is certainly important, and novel. This is complementary to the fact that there are generally few use-cases applying an XAI method on a real-world problem, most of them centered around the Global North. A recent survey on XAI projects centered around the Global South [21] showed that while the body of work in this field is growing rapidly, only 16 of the surveyed papers relate to the Global South, with only one technical study that has similar focus and design as ours [13].

Our analysis showed that XAI methods can be useful to verify an ML model in practice. However, we would like to caution against using these tools blindly, and summarize the challenges that emerged during our work as follows:

**Pipeline Design** If the aim is to use an XAI method, data processing and choosing the model tasks are limited by the fact that most XAI methods focus on tabular, image or text data, and on classification problems. Adapting a problem to this could lead to a loss of information, and lower prediction accuracy. Thus, efforts should be made to provide explanations for other task and data types (such as LASTS [11] for time series data, and beyond).

**Domain Knowledge** While an isolated XAI method can be very useful for debugging purposes, domain knowledge is necessary to draw real-world connections, and eventually verify the model via the explanation. In our work, an example of such knowledge is the distribution of text message activity over Senegal. While domain knowledge needs to be available in the first place, it is usually external to the explanation. A direct integration into XAI methods via symbolic approaches could be therefore highly useful [4, 6].

Work presented in this paper relies on a static network. We initially kept a second version of the dataset in the form of time series. Details on these experiments, including the trained ML model and explanations based on LASTS [11] can be found in the appendix A.5. The time series data posed several challenges, among others, an high need of computing resources, few XAI approaches to use and compare, and a considerably higher amount of domain knowledge needed for the interpretation of the data. For these reasons, we decided to continue working only on the network data, and leave the time series data for future work.

## Acknowledgments and Disclosure of Funding

This work has received funding from the European Union's Horizon 2020 research and innovation programme under Marie Sklodowska-Curie Actions (grant agreement number 860630) for the project "NoBIAS - Artificial Intelligence without Bias" (*nobias-project.eu*). This work reflects only the authors' views and the European Research Executive Agency (REA) is not responsible for any use that may be made of the information it contains.

## Authors' contributions

*Laura State:* conceptualization, experiments, paper draft, writing and editing *Hadrien Salat:* data preparation, paper co-writing and editing *Stefania Rubrichi:* data curation, paper reviewing, co-supervision *Zbigniew Smoreda:* data curation, paper reviewing

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

# A Appendix

## A.1 Data Distribution

Figure 4 shows the distribution of available call data (left panel) and text message data (right panel). While both data are more dense in the Western part of Senegal, text message data are specifically sparse in the Eastern part of the country.

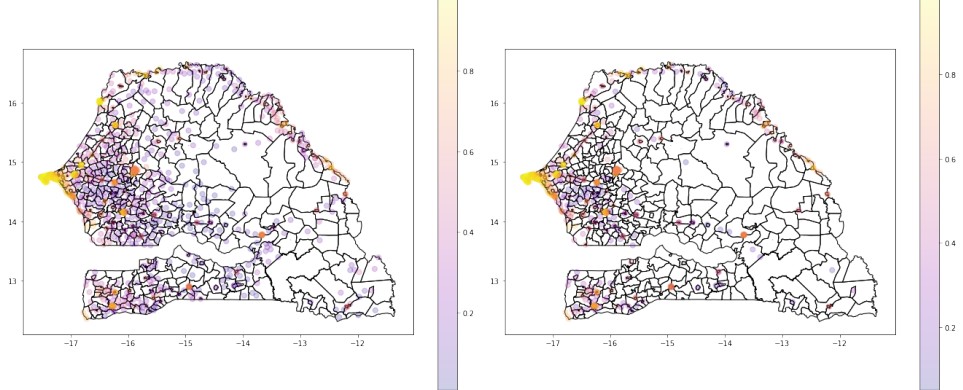

Figure 4: Spatial distribution of available data. Left panel: call data, right panel: text message data. Colored points reference to cell tower locations. Call data based on CN. Plots based on time series data, outgoing.

## A.2 Data Preparation

We build a directed communication network from the data (one per data type). Edges are formed based on the activity in the network. Per cell tower (outgoing site in the original dataset) we aggregate over the receiving tower (incoming site in the original dataset) and sum over respective events, for example the number of calls between those two towers. We thereby do not take care of the time stamp. The total number of events per connection, and in the full year 2013, determines the weight of the edge. This will lead to a full matrix ($M_{events}$). It has elements on its main diagonal, as calls/messages appear also within a Voronoi cell. Also, it is not symmetric as the number of outgoing and incoming calls/messages between two cell towers is generally not the same.

We repeat this network construction for all three types of data (call numbers, call length and message data, ). We extract six features per cell tower per network, and aggregate them, resulting in a dataset of dimensionality $1587 \times 18$.

## A.3 Additional Plot SHAP

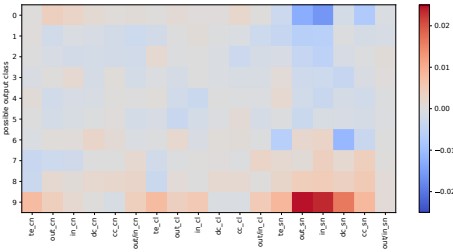

Figure 5: Average contribution of a feature w.r.t. the possible output class.

Figure 5 show the average feature contribution w.r.t. to all possible output classes.

Why do we observe negative contributions for low classes? For a single data instance, features based on text message data generally pushes the model towards the predicted class (high positive contribution) but at the same time away from the other classes (high negative contribution), together they form the set of possible output classes. This effect is more prominent for data instances from higher classes, thus positive (negative) contributions are particular high for high (low) classes.

Opposed, in figure 2, right, for a better comparability with LIME, we only plotted contribution w.r.t. to the predicted class, thus only positive contributions appear.

## A.4 Disaggregated Explanations

Disaggregated explanations, based on SHAP and w.r.t. possible output classes are shown in figure 6.

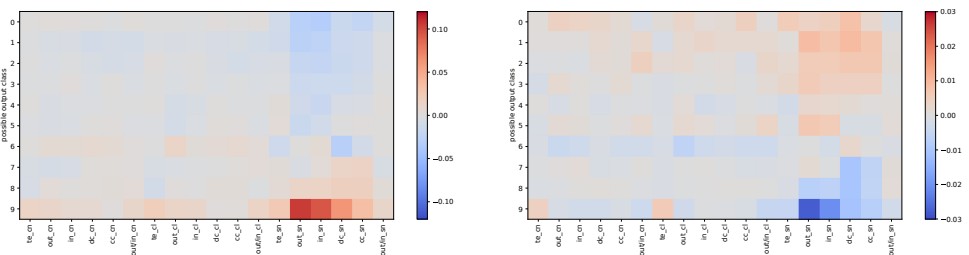

Figure 6: Explanations by sub-population, w.r.t. possible output classes as provided by SHAP: for urban regions (left), and rural regions (right).

## A.5 Time Series Data

In this section, we briefly describe the work on time series data.

### A.5.1 Data Processing

A time series $S = s_1..s_T$ consists of $T = 24 \times 12$ ordered data points, each being the monthly average of the aggregated number of events per hour, such that $s_t, t \in 1..24$ represent the monthly average of the aggregated number of events per hour in January ("daily activity curve" for January), etc. Events are separated by direction (incoming or outgoing). Thus, per cell tower, we create *six* time series. We refer to the TS dataset based on number of calls as CN, based on length of calls as CL and based on number of text messages as SN, and use out/in for outgoing/incoming activity, respectively. We standardize each of the time series separately by applying the min-max scaler as provided by sklearn.

Data labeling and subsampling applies as above. Data partitioning follows [11].

We find that text message data is heavily imbalanced, and is smaller than the other datasets. Thus, we exclude this data from the time series analysis.

The variational autoencoder that is used in the explanation as displayed below, is trained for $k = 50$ dimensions and over $e = 500$ epochs. We used the "out CL" data and model for explanations as it provides the smallest $MAE$.

### A.5.2 Classifications

To classify based on time series data, we use ROCKET (RandOm Convolutional KErnel Transform) [8], a method based on random convolutional kernels for feature extraction and linear classification. Results are displayed in table 3.

### A.5.3 Explanations

Figure 7 shows a sample explanation by LASTS [11]. Explanations are provided in visual form and as rule, the latter can be read off the plot. The time series belongs to class 4, i.e. has an electrification

Table 3: Classification results. The higher the accuracy the better, the lower the $MAE$ the better.

| data | $acc$ | $MAE$ | $MAE / MAE_{max}$ |
|---|---|---|---|
| out CN | 0.605 | 0.758 | 0.084 |
| out CL | 0.600 | 0.637 | 0.071 |
| in CN | 0.601 | 0.761 | 0.085 |
| in CL | 0.532 | 0.814 | 0.090 |

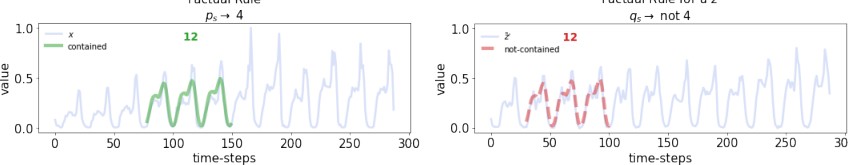

Figure 7: Local explanation by LASTS, shapelet-based, out CL data. Explained time series belongs to class 4, correctly classified by ML model. Left: factual rule, plotted against original time series, right: rule of opposite class, plotted against synthetically generated time series. Time steps in hours.

rate between $0.4$ and $0.5$, and is correctly classified by the model. In the left panel, the factual rule is plotted against the original time series. The number above the shapelet indicates its index. The rule reads as follows: "If shapelet no. 12 is contained in the time series, then it is classified as class 4." This is mirrored by the rule for instances belonging to the opposite class (here: class 0..3, 5..9): 'If shapelet no. 12 is not contained in the time series, then it is not classified as class 4.", displayed in figure 7, right, plotted against a synthetically generated time series from a class different to class 4.

