# OpenReview forum: "Explainability in Practice: Estimating Electrification Rates from Mobile Phone Data in Senegal"
_NeurIPS.cc/2022/Workshop/TSRML — TSRML2022_

### Official Review · Reviewer_GdtA · 2022-10-21
**Great motivation, would like to see more result details**

**Overall Rating:** 5

**Summary:**

This paper performs a practical investigation of the efficacy of explanation techniques. The authors focus on a case study of mobile phone data in Senegal and seek to train and explain a classifier that learns the correlation between phone data and stable electricity access for different regions of the country. The authors then compare the consistency between LIME and SHAP.

**Strengths:**

This is a well written paper that focuses on an important issue. We certainly do not have enough studies that evaluate XAI methods in practice.

The results are interesting and show that these local explanation methods can consistently identify important features.


**Weaknesses:**

The major weakness of this work, in my view, is that not enough time is spent detailing the results. While features such as number of outgoing text messages are clearly important, there are also  inconsistencies between the two charts. For example, the authors mention that LIME has negative values, while SHAP does not, but they do not explain why they think this difference occurs. Moreover, SHAP seems to have a higher variance in feature importance as compared to LIME (i.e., more LIME cells in the chart are close to the average color than in SHAP). I feel that this is  a valuable property for an explanation method, and would imply SHAP as a stronger practical alternative to LIME, but I do not see a strong comparison in the text. The authors provide the chart in Figure 2 and let it speak for itself. I don't feel that that is enough here.

Additionally, it felt like the work focused too much on setup and background details of the dataset. While this work seeks to validate XAI methods, 4/6 pages are setup, 1/6 pages talks about results, and 1/6 pages are limitations and conclusion. I feel like you could have removed some of the background information to provide more detail on the analysis.

I understand that there are limits to the number of classifiers that you can train, but it seems odd to me that the classifiers perform so poorly on rural data. It may be more helpful to train classifiers for rural and urban and explain those separately. Also, Is there a particular reason you limited the classifiers to DT, RF, and XGB? It would make more sense to me stop if XGB performed the best, but since RF has the highest accuracy, I would have liked to see a larger classifier comparison. Based on the small dataset size, it seems fairly trivial to grab a large number of different classifier types from  sklearn, retrain on each classifier, and include a table of performances in the appendix.

As a final point, this task seems simple enough that I would have liked to see a comparison to an inherently interpretable model, e.g. logistic regression.


**Overall Recommendation:**

I feel like the goal of this paper is interesting, but that the authors could have had much more detail. I felt that I had several questions that were not answered about their results.

**Review Confidence:**

4: The reviewer is confident but not absolutely certain that the evaluation is correct

---

### Official Review · Reviewer_D6zh · 2022-10-22
**Promising initial work on an important application**

**Overall Rating:** 7

**Summary:**

The authors conduct an experiment which seeks to first predict electrification rates in Senegal from mobile phone data, then confirm the validity of the model by appealing to explainability techniques. Specifically, the goal is to confirm the validity of an uninterpretable model (a random forest) by showing that it relies on features which are known to domain experts to be helpful in predicting the target variable (electrification rate). They test two popular model interpretation methods, LIME and SHAP, and confirm that the random forest tends to predict that the electrification rate is higher in areas with higher text messaging rates, which aligns with domain knowledge. In an orthogonal direction, the authors find that predictive models have worse accuracy in areas with lower population densities.

**Strengths:**

The problem itself is important. As the authors mention, collecting census or survey data can often be expensive and time consuming, but the knowledge that such data provides is important for policy decisions. This seems especially urgent for developing countries. Thus, techniques for gathering the required knowledge via more readily available data sources is valuable, and when this data will be used to make consequential decisions in the future, making sure that the model used to create it is a reliable one is also critical.

**Weaknesses:**

The goal of the paper/the meaning of "validating" the model wasn't clear to me at first. If my interpretation is correct, "validate" means confirm that the model relies on features which make sense in the context of domain knowledge, but I think it would be helpful to make this more explicit earlier in the paper.

A more thorough discussion of the results would also be helpful. It's interesting that both LIME and SHAP identify the same text message related features as important for predicting areas with high electrification rate, but it looks like none of the other classes have any consistent interpretation according to these methods. It seems like areas with low electrification (class 0) should have a large negative importance with respect to these features, for instance. Can you provide some more intuition/interpretation for the other classes?

Minor:
- Typo: "desgin" on line 37

**Overall Recommendation:**

This seems like a promising initial work with clear relevance to the workshop. ML applications for the benefit of the developing world seem especially important for the goal of social responsibility. I recommend accept.

**Review Confidence:**

3: The reviewer is fairly confident that the evaluation is correct

---

### Official Review · Reviewer_DKVb · 2022-10-23
**A useful case study of explanability methods in a real life scenario with potential warnings/recommendation**

**Overall Rating:** 6

**Summary:**

The authors train a model to predict electrification rate in Senegal. They use two explanability methods to determine that population density disproportionately affects the predictions and can misleadingly determine higher electrification rates in high population density areas i.e. urban areas. They point out one is advised to use explanability methods especially in the cases where such ML models are used to make policy decision.

**Strengths:**

The authors do a good job in motivating the problem setting. The recommendation to focus on the importance of explanable AI methods in cases where one needs to make policy decisions is a useful point well made.

**Weaknesses:**

The authors claim that " while we need a sufficiently high performance of our models to proceed, the focus of this paper is not on model performance". I wonder how much do the outcome of the experiments , i.e. the finding that the model is biased on population density would change in case of a better performing model.

Also do the result vary across different types of models (i.e. svm, neural nets etc)? These experiments would make the paper much more compelling.

**Overall Recommendation:**

The paper puts forward a compelling case to push for the usage of explanable AI techniques. This recommendation is based on an experiment on predicting electrification rates in senegal and how models can be biased towards features which eventually can lead to wrongly motivated policy decisions. Even though I believe the experiments are lacking in two main ways 1) how does change in accuracy change the resulting explanability outcome and 2) how does the choice of model affect the outcome; I feel the paper has done a good job in adding another datapoint in practical studies of explanable AI.

**Review Confidence:**

3: The reviewer is fairly confident that the evaluation is correct

---

### Decision · Program_Chairs · 2022-10-23

**Decision:**

Accept

**Comment:**

This submission is accepted based on its novel application of XAI methods in a concrete problem - estimating electrification rates. As both reviewers point out, the clarity of the submission and soundness of experimental evaluation should be improved - we strongly recommend the authors improve this part in the camera-ready version.